# Small Heat Shock Proteins Collaborate with FAIM to Prevent Accumulation of Misfolded Protein Aggregates

**DOI:** 10.3390/ijms231911841

**Published:** 2022-10-06

**Authors:** Hiroaki Kaku, Allison R. Balaj, Thomas L. Rothstein

**Affiliations:** Department of Investigative Medicine, Western Michigan University Homer Stryker M.D. School of Medicine, Kalamazoo, MI 49007, USA

**Keywords:** fas apoptosis inhibitory molecule (FAIM), small heat shock proteins (sHSPs), protein aggregates

## Abstract

Cells and tissues are continuously subject to environmental insults, such as heat shock and oxidative stress, which cause the accumulation of cytotoxic, aggregated proteins. We previously found that Fas Apoptosis Inhibitory Molecule (FAIM) protects cells from stress-induced cell death by preventing abnormal generation of protein aggregates similar to the effect of small heat shock proteins (HSPs). Protein aggregates are often associated with neurodegenerative diseases, including Alzheimer’s disease (AD). In this study, we sought to determine how FAIM protein dynamics change during cellular stress and how FAIM prevents the formation of amyloid-β aggregates/fibrils, one of the pathological hallmarks of AD. Here, we found that the majority of FAIM protein shifts to the detergent-insoluble fraction in response to cellular stress. A similar shift to the insoluble fraction was also observed in small heat shock protein (sHSP) family molecules, such as HSP27, after stress. We further demonstrate that FAIM is recruited to sHSP-containing complexes after cellular stress induction. These data suggest that FAIM might prevent protein aggregation in concert with sHSPs. In fact, we observed the additional effect of FAIM and HSP27 on the prevention of protein aggregates using an in vitro amyloid-β aggregation model system. Our work provides new insights into the interrelationships among FAIM, sHSPs, and amyloid-β aggregation.

## 1. Introduction

Cells are consistently exposed to either intracellular or environmental stressful stimuli, such as oxidative or heat stress, which causes cellular damage, including oligomerization/aggregation of dysfunctional/disordered proteins. Cells respond to stress in a variety of ways. Initially, cells try to recover protein homeostasis (hereafter: proteostasis) by upregulating and activating heat shock protein (HSP) family molecules, in addition to proteasomal degradation and autophagic elimination. However, if the noxious stimulus is overwhelming and unsolved, then cells activate death signaling pathways to eliminate damaged cells. The ability to maintain protein homeostasis within cells is essential for cellular health, especially in post-mitotic cells that cannot dilute the ill-effects of misfolded, unfolded, or aggregated proteins through cell division.

HSP family molecules act as molecular chaperones to maintain proteostasis in cells by promoting the folding of newly synthesized proteins or by preventing protein misfolding/aggregation induced by cellular stress. HSPs have been organized into seven major classes based on molecular size, including HSP10, small heat shock proteins (sHSPs), HSP40, HSP60, HSP70, HSP90, and HSP110 [1].

We previously found that fas apoptosis inhibitory molecule (FAIM) protects cells from stress induced cell death by preventing protein aggregation similar to HSPs [2]. Moreover, FAIM possesses a chaperone-like activity given that it prevents the de novo formation of aggregates from SOD1-G93A and amyloid-β in cell-free systems [3,4]. FAIM was originally cloned as a FAS antagonist in mouse primary B lymphocytes [5]. A subsequent study identified the alternatively spliced form, termed FAIM-Long (L), which has 22 additional amino acids at the N-terminus [6]. Thus, the originally identified FAIM was renamed FAIM-Short (S) [6]. FAIM-L is expressed almost exclusively in the brain and testes, whereas FAIM-S is ubiquitously expressed [6]. However, whether FAIM expression is enhanced in the event of cellular stress similar to some HSPs and how FAIM prevents protein aggregation still remain unknown. Although we showed FAIM is recruited to a complex containing ubiquitinated protein aggregates after stress induction (Kaku and Rothstein, 2020), its mechanism of action remains unknown as there is still a lack of recognized effector/binding motifs or even partial sequence homology of FAIM with any other protein.

In order to obtain some clues of how FAIM functions to protect cells from stress-induced cell death, we performed an in silico analysis to find candidates for binding partner(s) of FAIM using the WormBase (https://wormbase.org, accessed on 4 March 2014), that contains the identity of interacting *C. elegans* proteins recognized by yeast two hybrid screening [7]. According to the database, D1081.6 and hsp-25 proteins are interactors with the *C. elegans* FAIM ortholog, C44B11.1. Although the *D1081.6* gene is not conserved in mammals, the *hsp-25* gene is homologous to mammalian *sHSP* genes, including human heat shock protein 27 (HSP27) (also termed HSPB1) (Appendix A). However, whether this interaction is conserved in mammalian cells remains unexplored.

In this study, we seek to investigate how FAIM protein expression/dynamics change in the event of cellular stress induction, and whether human FAIM protein interacts with human sHSPs.

## 2. Results 

### 2.1. FAIM mRNA Expression Is Not Up-Regulated after Cellular Stress Induction in HeLa Cells

We previously showed that FAIM protein possesses heat shock protein (HSP)-like chaperone activity given that FAIM prevents amyloid-β and SOD1 aggregation in an in vitro cell-free system [3,4]. Some HSPs respond to stress conditions by upregulating their expression [8]. We examined FAIM expression in HeLa cells to determine if its expression is upregulated by heat shock, similar to some HSPs. We found *FAIM* mRNA expression was not increased under stress conditions, in contrast to *HSPs* such as *HSP27/HSPB1*, *αB-crystallin/HSPB5*, *HSP70 A1A* and *HSP90 AA1* which were increased (Figure 1).

### 2.2. FAIM Protein Shifts to the Detergent-Insoluble Fraction after Stress Induction

Further, we tested FAIM protein expression levels with or without cellular stress induction in HeLa cells. FAIM protein expression levels were actually decreased in the RIPA lysis buffer soluble fraction upon heat shock and oxidative stress induced by menadione (Figure 2). However, additional analysis determined that FAIM protein had shifted to the detergent-insoluble fraction in response to cellular stress (Figure 2), which was especially noticeable after heat shock. To exclude the possibility that FAIM protein becomes an insoluble precipitate due to denaturation caused by heat shock, HeLa cell lysates and live HeLa cells were incubated at 43 °C. FAIM protein from HeLa cell lysates stayed in the detergent soluble fraction, suggesting that FAIM protein is actively translocated into the insoluble fraction in the live cells (Appendix A). Furthermore, recombinant FAIM protein was also still soluble after incubation at 43 °C or 70 °C although soluble multimer FAIM protein bands were observed at 70 °C, indicating that FAIM protein itself is resistant to denaturation caused by high temperature (Appendix A). Thus, we conclude that FAIM protein is translocated into the insoluble fraction after heat shock in a cell-intrinsic manner.

To validate these results using a different extraction method, we separated HeLa proteins into 4 fractions—cytosol, membrane/organelle, nuclear, and cytoskeletal/detergent-insoluble—after mild heat stress. We found that the majority of FAIM protein migrated to the cytoskeletal/detergent insoluble fraction (Figure 3). Large HSPs such as HSP90, HSP60, and HSP40 maintained their original subcellular distribution after heat stress, whereas HSP27 showed a marked shift to the cytoskeletal/detergent insoluble fraction (Figure 3). Proteins from HLE B-3 cells, which abundantly express other sHSPs such as αA-(also termed HSPB4) and αB-crystallins (also termed HSPB5) in addition to HSP27 [9,10], were similarly analyzed with respect to subcellular distribution before and after heat stress. Here, again, FAIM, HSP27 and related crystallin proteins migrated to the cytoskeletal/detergent insoluble fraction in response to heat stress, unlike other proteins (Appendix A). Thus, the bulk of FAIM protein migrates to the detergent-insoluble fraction when cells are exposed to stress, as do sHSPs [11,12]. These data strongly suggest that FAIM protein behaves similarly to sHSPs after the induction of cellular stress.

### 2.3. FAIM Is Recruited to the Complex Containing sHSPs after Cellular Stress Induction

To investigate the possibility that FAIM directly associates with HSP27-containing complexes, we carried out co-immunoprecipitation (co-IP) followed by Western blotting. A FLAG-FAIM-S expression vector was transfected into FAIM-KO HeLa cells and then FLAG-FAIM-S was immunoprecipitated with FLAG-binding resin and HSP27 protein was visualized with an anti-HSP27 antibody. When cells were exposed to cellular stress, HSP27, but not other HSPs, co-immunoprecipitated with FAIM-S (Figure 4). Co-IP data was further verified by in situ proximity ligation assay (PLA), in which protein–protein interaction was visualized in situ [13]. Theoretically, protein–protein interaction occurring in the detergent-insoluble fraction, as well as the detergent-soluble fraction, can be detected by in situ PLA [14], whereas protein–protein interaction detected by co-IP assay is theoretically limited only to detergent-soluble proteins. Whereas PLA did not show any signal in the absence of FLAG-FAIM-S (Figure 4B), we observed strong PLA signals with diffuse, widespread distribution in the cytoplasm in the presence of FLAG-FAIM-S upon stress induction, indicating close proximity between FAIM-S and HSP27 after stress (Figure 4C).

To extend the results to other sHSPs, HLE B-3 cells, which abundantly express HSP27, αA- and αB-crystallins, were utilized for co-IP. Cells were similarly treated and harvested. We found that αA- and αB-crystallins in addition to HSP27 were co-immunoprecipitated with FAIM-S after cellular stress induction (Appendix A). These data indicate that FAIM is recruited to a complex containing sHSPs or FAIM directly interacts with sHSPs, in response to cellular stress induction before becoming detergent-insoluble.

### 2.4. Recombinant FAIM Inhibits β-Amyloid Aggregation/Fibrillization in Collaboration with HSP27 in an In Vitro Cell-Free System

FAIM-S and sHSPs are known to inhibit β-amyloid fibrillization/aggregation in vitro [4,15,16]. To examine whether FAIM-S and HSP27 have an additive effect on the prevention of aggregation, we utilized the in vitro β-amyloid fibrillization/aggregation model. Sub-optimal doses of recombinant FAIM and β-amyloid monomer without (Figure 5A) or with (Figure 5B) HSP27 were mixed, and aggregation status was monitored in real-time using ThT fluorescence intensity. We found that β-amyloid aggregation was completely abrogated in the presence of both FAIM-S and HSP27 whereas aggregation was only partially inhibited (or slowed) in the presence of either FAIM-S and HSP27 alone (Figure 5A,B). Furthermore, sub-optimal doses of recombinant HSP27 without (Figure 5C) or with (Figure 5D) FAIM-S were also monitored, and β-amyloid aggregation was completely inhibited in the presence of both FAIM-S and HSP27 in this condition (Figure 5C,D). These data suggest that FAIM-S prevents aggregation in collaboration with HSP27.

## 3. Discussion

We previously reported that FAIM, similar to HSP27, prevents the formation of cytotoxic protein aggregates after cellular stress, which protects cells from loss of viability [2,17,18]. Subsequent studies demonstrated that FAIM can antagonize SOD1 and amyloid-β aggregation using an in vitro cell-free model, indicating that FAIM may have chaperone-like activity similar to HSPs [3,4]. The work described herein finds a FAIM protein shift (or translocation) after stress similar to sHSPs. Thus, FAIM-S protein seems to mimic the response of HSP27 in cells exposed to stress. Importantly, our work identifies a FAIM-sHSP interaction specifically after induction of cellular stress and indicates that the interaction can prevent protein aggregates more efficiently than FAIM or HSP27 alone. As we previously showed that FAIM is recruited to a complex containing ubiquitinated protein aggregates after stress induction [2], we hypothesize that both FAIM and HSP27 are recruited to the same complex before ubiquitinated proteins become insoluble aggregates to achieve more efficient prevention of protein aggregation upon cellular stress.

HSP27, αA-crystallin, and αB-crystallin belong to the sHSP gene superfamily. There are 10 human sHSP genes, including *HSPB1* coding for HSP27, *HSPB4*/*CRYAA* coding for αA-crystallin, and *HSPB5*/*CRYAB* coding for αB-crystallin [19,20]. Most sHSPs have chaperone activity and function by preventing protein aggregation induced by various stress conditions in an ATP-independent manner, which renders stress tolerance and prevents apoptosis [21,22,23]. sHSPs have a highly conserved stretch of 80–100 amino acids in their C-terminal domains called the “α-crystallin domain” that is flanked by a less conserved N-terminal domain and a C-terminal extension. Some members of the sHSP family, such as HSP27, αA- and αB-crystallin, can form large oligomers (up to 900 kDa) resulting from inter-sHSP interaction through a subunit exchange process, enhancing their chaperone activity [24,25,26,27]. However, whether FAIM is incorporated into the large oligomer still remains unexplored.

Although we have found that FAIM closely resembles sHSPs in many respects during cellular stress, no homologous regions or shared domains have been identified, which implies that they evolved independently. We found some other similarities between FAIM and sHSPs in previously reported literature. First, overexpression of FAIM in the PC12 neuron-like cell line [28] promotes nerve growth factor (NGF)-induced neurite outgrowth in vitro similar to overexpression of HSP27 in dorsal root ganglion neurons [29]. Second, FAS-mediated apoptosis can be inhibited by overexpression of FAIM in B lymphocytes [5], in PC12 cells [30], and in HEK293T cells and cortical neurons [31], or by overexpression of HSP27 in L929 cells [32]. Third, NF-κB activation is enhanced by overexpression of FAIM in NGF-treated PC12 cells [28] and in CD40-stimulated B lymphocytes [33] whereas overexpression of HSP27 enhances NF-κB activation in TNF-α-treated U937 cells and MEF cells [34]. It is possible some of these effects could be artifacts of overexpression due to protein/gene dosage imbalances that alter biological outcomes rather than from direct biological effects of FAIM or HSP27 [35,36,37]. However, although a range of cells are involved, these effects suggest functional similarities between FAIM and HSP27.

Furthermore, we found structural and biochemical similarities, in addition to functional similarities by overexpression, that suggest FAIM and sHSPs contribute to related activities in cells, including the prevention of protein aggregation in an ATP-independent manner. First, the predicted molecular weight of monomer FAIM ranges from 17–43 kDa throughout the evolution of holozoans while sHSPs have a similar monomeric molecular weight range of 15–40 kDa [38]. Second, three-dimensional protein analysis by NMR reveals that FAIM contains a rare compact non-interleaved seven stranded β-sandwich structure with an anti-parallel β-sheet in the C-terminal region and a highly disordered N-terminal region [31,39]. Highly disordered protein regions have been suggested to be important in binding aggregation-prone targets [40]. Strikingly, similar structural features are observed in sHSPs [27,41]. Third, amino acid composition analysis of FAIM protein sequences from choanoflagellates to human shows significant underrepresentation of cysteine residues (only 1.4%), which is also true of sHSPs [42]. A reduced number of cysteines may be important to prevent unwanted crosslinking under oxidative conditions, enabling FAIM and sHSPs to resist denaturation and perform their common function to prevent protein aggregation induced by cellular oxidative stress [42]. However, despite these many similarities between FAIM and sHSPs, FAIM is not an sHSP. FAIM is not homologous with sHSPs and contains no β-crystallin domain seen in most sHSPs.

In sum, we conclude that FAIM can interact with sHSPs and in so doing prevents the formation of protein aggregates more efficiently. It has been reported that expression levels of FAIM are impaired in the hippocampus of Alzheimer’s disease patients [43] whereas the expression levels of some sHSPs are elevated in AD brains [44,45]. These observations led us to speculate that elevated expression of sHSPs might be a compensatory response to low FAIM expression in AD that antagonizes amyloid-β aggregates, albeit less efficiently than with FAIM. Although it has been suggested that sHSPs may be useful in treating neurodegenerative diseases by reducing or preventing proteins aggregates, the combined therapy of FAIM and sHSPs could be more effective than the administration of a single molecule. To combat devasting neurodegenerative diseases, it would be important in the future to examine whether and to what extent FAIM-sHSP interaction plays a role in preventing protein aggregation in vivo, and how FAIM interacts with sHSPs.

## 4. Materials and Methods

### 4.1. Reagents and Antibodies

Goat anti-HSP27 (M-20), and mouse anti-αA-crystallin (B-2) antibodies were obtained from Santa Cruz Biotechnology (Dallas, TX, USA). Rabbit anti-vimentin, rabbit anti-histone H3, rabbit anti-MEK1/2, rabbit anti-HSP40, rabbit anti-HSP60, rabbit anti-HSP70, rabbit anti-HSP90, rabbit anti-AIF, goat anti-rabbit IgG-HRP-linked and horse anti-mouse IgG-HRP-linked antibodies were obtained from Cell Signaling Technology (Danvers, MA, USA). Mouse anti-FLAG (M2) and mouse anti-β-actin antibodies were obtained from Millipore Sigma (St. Louis, MO, USA). Rabbit αB-crystallin antibody was obtained from Enzo Life Sciences (Farmingdale, NY, USA). Affinity purified anti-FAIM antibody was obtained from rabbits immunized with CYIKAVSSRKRKEGIIHTLI peptide (located near the C-terminal region of FAIM) as previously described [33,46].

### 4.2. Gene Expression Analysis by qPCR

Gene expression was assayed by real-time PCR as previously described [46]. Briefly, RNA was prepared from cells using the RNeasy mini kit (Qiagen, Hilden, Germany), according to the manufacturer’s instructions. cDNA was prepared using iScript reverse transcription supermix (Bio-Rad, Hercules, CA, USA). Gene expression was then measured by real-time PCR using iTaq SYBR Green (Bio-Rad) and normalized with GAPDH. Primer sequences are shown in Appendix A.

### 4.3. In Vitro Cellular Stress Induction

To induce mild heat shock, cells in culture dishes were incubated in a water bath at 43 °C for the indicated period [47]. In some experiments, cells were recovered at 37 °C after heat stress induction at 43 °C for 2 h as previously described [47]. To induce oxidative stress, menadione (MN) (Millipore Sigma), dissolved in DMSO at 100 mM, was added to the medium at the indicated final concentration for 1 h. In oxidative stress experiments where cells were harvested at time points beyond 1 h, cells were washed once with medium and fresh medium (without menadione) was added into the cell culture as previously described [48].

### 4.4. Cell Culture and Transfection

HeLa, and HLE B-3 cell lines were obtained from the American Type Culture Collection (ATCC). FAIM-KO HeLa and FAIM-KO HLE B-3 cells were generated with CRISPR/Cas9 as previously described [3]. HeLa cells were cultured in DMEM medium (Corning) whereas HLE B-3 cells were cultured in EMEM (Corning). Both DMEM and EMEM contained 10% FBS, 15 mM HEPES, 2 mM L-glutamine and 0.1 mg/mL penicillin and streptomycin. Transfection was performed using Lipofectamine 2000 for HLE B-3 cells or Lipofectamine 3000 for HeLa cells, according to the manufacturer’s instructions (Themo Fisher Scientific, Foster City, CA, USA).

### 4.5. His-Tag Recombinant Protein Production

His-tag protein expression vectors were constructed using pTrcHis TA vector (Themo Fisher Scientific) according to the manufacturer’s instructions as previously described [3]. In brief, PCR amplified target genes were TA-cloned into the pTrcHis TA vector (Themo Fisher Scientific) and inserted DNA was verified by sequencing (Genewiz, South Plainfield, NJ, USA). Proteins were expressed in TOP10 competent cells (Themo Fisher Scientific) with IPTG (Themo Fisher Scientific) at 1 mM for 2.5 h at 37 °C, and were purified using a Nuvia IMAC Nickel-charged column (Bio-Rad) on an NGC Quest chromatography system (Bio-Rad). Proteins underwent dialysis against PBS using Slide-A-Lyzer Dialysis Cassettes, 10 K MWCO (Themo Fisher Scientific).

### 4.6. Construction of FLAG-Tag FAIM-S Expression Vector

The FLAG-FAIM-S expression vector was constructed using pCMV-(DYKDDDDK)-N (FLAG-N) vector (Takara). Human *FAIM-S* cDNA was cloned into EcoRⅠ and KpnⅠ sites in the vector. Primers for gene cloning are shown in Appendix A. The insert was verified by sequencing (Genewiz).

### 4.7. Thioflavin T Fluorescence Assay

Fibril/aggregate formation of 5 μM Aβ (1–42) (Aβ42) (Anaspec, Fremont, CA, USA), assembled in a 384-well clear bottom plate (Corning), was assessed by 50 μM Thioflavin T (ThT) (Millipore Sigma) fluorescence using a Synergy Neo2 Multi-Mode Microplate Reader (Bio-Tek, Winooski, VT, USA). Reader temperature was set at 37 °C with continuous shaking between reads. ThT fluorescence intensity was measured using an excitation wavelength of 440 nm and an emission of 482 nm. PMT gain was set at 80. Fluorescence measurements were made from the bottom of the plate, with the top being sealed with an adhesive plate sealer to prevent evaporation.

### 4.8. Western Blotting

Cells were washed twice with PBS and lysed in RIPA lysis buffer (1% Nonidet P-40, 0.5% sodium deoxycholate, 0.1% SDS, 150 mM NaCl, 50 mM Tris-HCl (pH 8.0), 2 mM EDTA) containing supplements of 2 mM Na_3_VO_4,_ 20 mM NaF, and a protease inhibitor cocktail (Millipore Sigma) for 30 min on ice. Lysates were clarified by centrifugation at 21,100× *g* for 10 min. Supernatants were used as RIPA-soluble fractions. The insoluble-pellets (the RIPA-insoluble fractions) were washed twice with RIPA buffer and proteins were extracted in 8 M urea in TBS. In some experiments, protein lysates were separated into 4 subcellular fractions (cytosolic, membrane/organelle, nucleic, and cytoskeletal/insoluble fractions) using ProteoExtract Subcellular Proteome Extraction Kit (Millipore Sigma) according to the manufacturer’s instructions. Protein concentration was determined using the 660 nm Protein Assay Reagent (Themo Fisher Scientific). Protein samples in 1 × Laemmli buffer with 2-ME were boiled for 5 min. Equal amounts of protein for each condition were subjected to SDS-PAGE followed by immunoblotting.

### 4.9. Immunoprecipitation

Cells expressing FLAG-tag proteins were lysed in RIPA buffer containing 2 mM Na_3_VO_4,_ 50 mM NaF, and protease inhibitor cocktail for 30 min on ice. Lysates were clarified by centrifugation at 21,100× *g* for 10 min. Equal amounts of protein for each supernatant were mixed with anti-FLAG M2 Magnetic Beads (Millipore Sigma) and incubated at 4 °C under gentle rotation for 2 h. Beads were washed with lysis buffer four times and FLAG-tag proteins were eluted with 100 μg/mL 3 × FLAG peptide (Millipore Sigma) two times. Eluates were pooled and Western blotting was performed to detect FLAG-FAIM-S-interacting proteins.

### 4.10. In Situ Proximity Ligation Assay (PLA)

Cells were cultured for 24 h on poly-L-lysine coated coverslips (Corning) in 24-well plates. Cells with or without cellular stress induction were fixed and permeabilized with ice-cold 100% methanol. PLA reaction was carried out according to the manufacturer’s instructions using Duolink in situ Detection Reagents (Millipore Sigma). Briefly, coverslips were blocked using Duolink Blocking Solution (Millipore Sigma) for 1 h at 37 °C, followed by incubation with mouse anti-FLAG and goat anti-HSP27 antibodies overnight at 4 °C. Coverslips were washed by 1 × Wash Buffer (Millipore Sigma) three times the next day, and were subsequently incubated with the PLA probes (PLUS-probe-coupled anti-mouse and MINUS-probe-coupled anti-goat antibodies) (Millipore Sigma). Duolink In Situ Detection Reagents Orange (Millipore Sigma) were used for ligation and signal amplification. ProLong Gold Antifade Reagent with DAPI (Cell Signaling Technology) was used to stain nuclei and to prevent the fading of fluorescence. In this procedure, bright fluorescent dots were observed when two probes are in close proximity. Fluorescence signals were visualized with a Nikon A1R^+^ confocal microscope (Nikon, Melville, NY, USA).

## Figures and Tables

**Figure 1 ijms-23-11841-f001:**
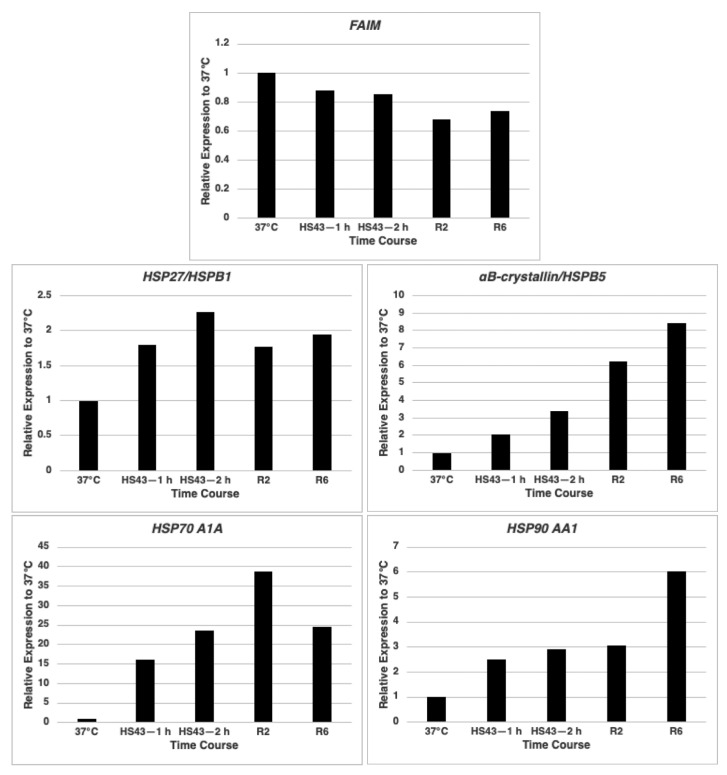
*FAIM* mRNA expression is not increased during cellular stress induction. *FAIM* mRNA expression levels during heat shock conditions were analyzed by qPCR. Primers for *HSPB1*, *HSPB5 (CRYAB)*, *HSP70 A1A* or *HSP90 AA1* were also used as positive controls of heat-shock-induced genes. R2; recovery at 37 °C for 2 h after heat shock at 43 °C for 2 h. R6; recovery at 37 °C for 6 h after heat shock at 43 °C for 2 h. Similar results were obtained from at least 3 independent experiments.

**Figure 2 ijms-23-11841-f002:**
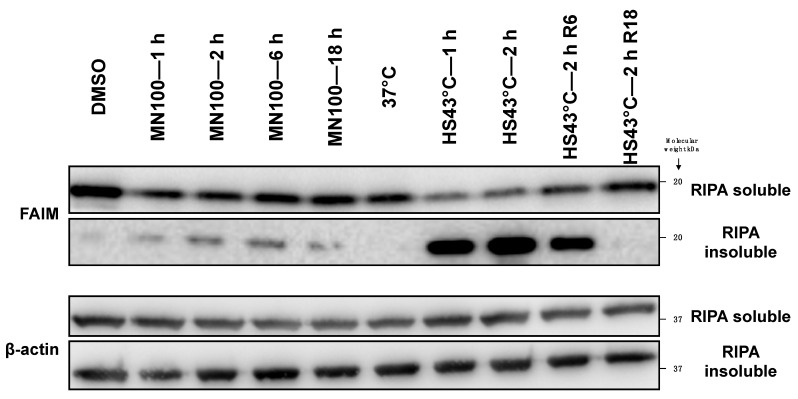
FAIM protein shifts to the detergent-insoluble fraction after stress induction. HeLa cells were exposed to heat shock (HS) at 43 °C for 1 h or for 2 h and were incubated at 37 °C for 6 or 18 h recovery (R6 and R18) after incubation at 43 °C for 2 h, as indicated. HeLa cells were also subjected to oxidative stress by treatment with 100 μM menadione (MN) for the indicated times (vehicle control; DMSO for 18 h). After stress induction, cells were harvested, soluble proteins were isolated using RIPA buffer and RIPA buffer-insoluble proteins were extracted. Equal amounts of protein for each fraction were analyzed by Western blotting for FAIM, and for actin as a loading control. Data representative of 3 independent experiments are shown.

**Figure 3 ijms-23-11841-f003:**
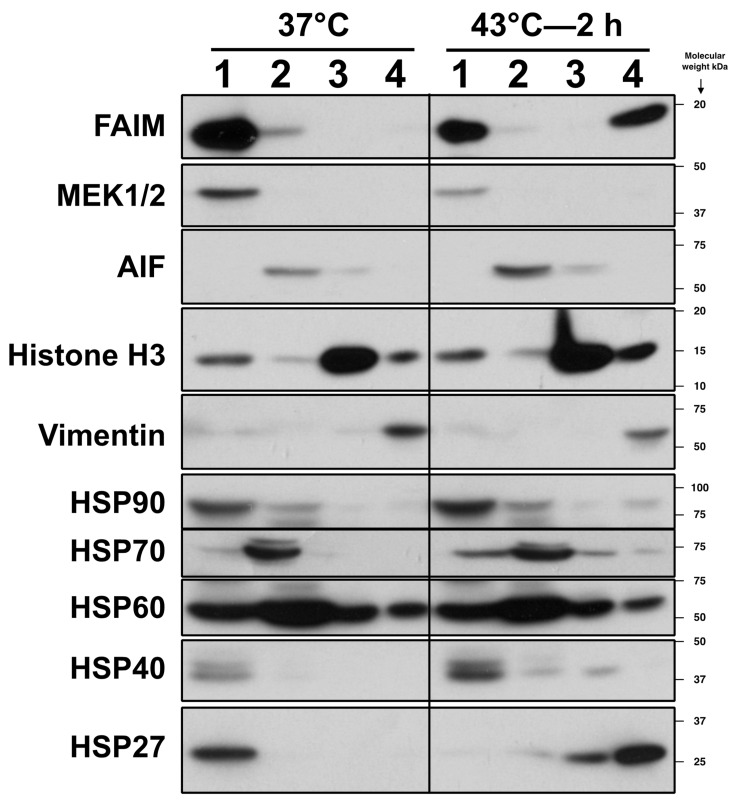
FAIM protein shifts to the vimentin positive, cytoskeletal/detergent-insoluble fraction after stress induction. HeLa cells were incubated at 37 °C, or were exposed to heat shock (43 °C) for 2 h, as indicated. Cells were then harvested, and proteins were divided into 4 fractions, (**1**); cytosol (MEK1/2-containing), (**2**); membrane/organelle (AIF-containing), (**3**); nucleus (histone H3-containing) and (**4**); cytoskeleton/insoluble (vimentin-containing). Equal amounts of protein were analyzed by Western blotting. Representative data are shown. Similar results were obtained from 3 independent experiments.

**Figure 4 ijms-23-11841-f004:**
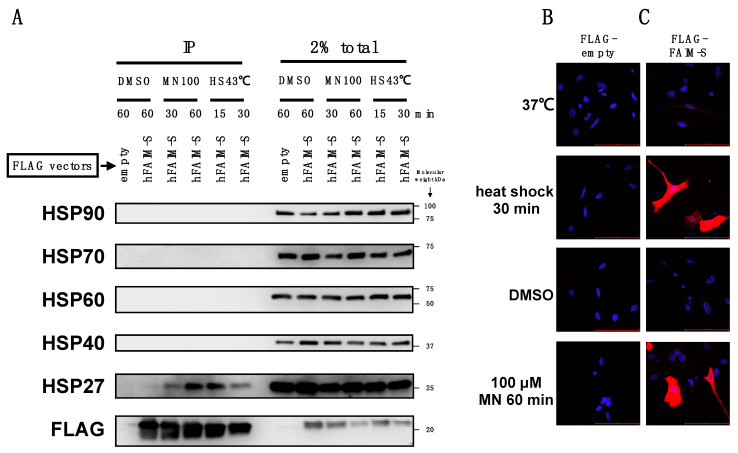
FAIM associates with HSP27 upon cellular stress. (**A**). FLAG-tagged FAIM-S was transiently transfected into FAIM-KO HeLa cells. Two days after the transfection, cells were harvested with or without stress induction provided by incubation with menadione (MN) at 100 μM or heat shock (HS) at 43 °C for the indicated times, and lysed in RIPA buffer. Samples were immunoprecipitated with anti-FLAG and then Western blotted for HSP proteins. (**B**,**C**). FAIM-KO HeLa cells transiently transfected with FLAG-tagged vectors were incubated at 43 °C for 30 min, with the DMSO diluent for menadione for 60 min, and with menadione at 100 μM (MN) for 60 min. Cells were then fixed, permeabilized, and a PLA reaction was performed. Red dots indicate positive PLA signals. The cell nuclei were stained blue with DAPI. Scale bar, 100 μm. Similar results were obtained from at least 3 independent experiments.

**Figure 5 ijms-23-11841-f005:**
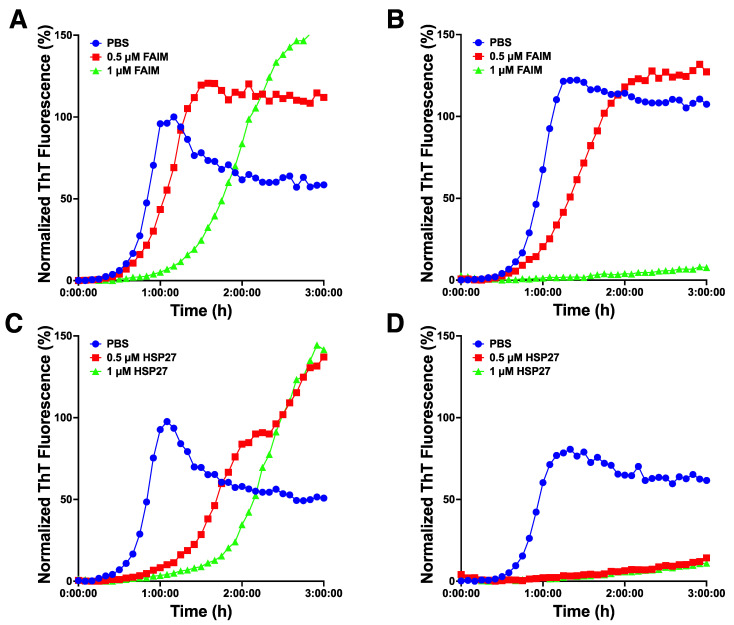
Recombinant FAIM-S protein suppresses amyloid-β fibrillization/aggregation in conjunction with HSP27 in a cell-free system. Spontaneous fibrillization of amyloid-β (5 μM) in vitro was monitored by ThT assay over a period of 3 h with the indicated concentration of recombinant FAIM-S in the absence (**A**) or presence (**B**) of 0.25 μM HSP27. ThT assay was also carried out with different concentrations of HSP27 in the absence (**C**) or presence (**D**) of 0.25 μM FAIM-S. ThT fluorescence was recorded every 5 min. Background ThT fluorescence is subtracted from sample readings at each time point, and is normalized to 100% of the maximum value in the PBS control. Representative data from 3 independent experiments are shown.

## Data Availability

All data supporting the findings of this study are available within the paper and within its Appendix A published online.

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
