# Peer review of "Small Heat Shock Proteins Collaborate with FAIM to Prevent Accumulation of Misfolded Protein Aggregates"

_ijms, 2022, doi:10.3390/ijms231911841_

Round 1
Reviewer 1 Report
In this manuscript the authors try to demonstrate a physical interaction between HSP27 and FAIM in preventing abnormal aggregation of protein upon heat stress.
However, it is not clear what small HSPs the authors talk about, because in there experiment they were using either a/b crystallin, HSP27, HSBP5.
Hence, the Title: “Additional effect” should be used instead of “synergic” because small HSPs and FAIM have probably target different substrates.
Questions:
Is HSP27 like FAIM block de novo formation of aggregates upon HS?
The sequence alignment of Hsp25 between C. elegans and human should be shown in supplementary.
Is FAIM upregulated in C. elegans upon stress?
Nomenclature must be unified HSBP5 = HSP27 ? small HSPs…
Figure 2: Why FAIM aggregates disappeared at 2h upon HS, is there re-solubilization of FAIM at 2H-R18 upon heat shock or lost upon cell division?
Figure 3 and supplementary Fig 3: is there an explanation why the two figures show a WB with a different distribution of HSP27?
Figure 4A (and supp. Fig 2) IP: Why there is a decrease in HSP27-FLAG-FAIM interaction at 30 min at 43°C, while FAIM is still aggregated after 2H at 43°C in the Figure 1?
The utilization of menadione (a ROS inducer) is not explained nor justified.
HSP27 is highly expressed but only a small fraction is coprecipitated with FAIM-FLAG. This may be relevant of an indirect interaction (via some common targets).
Figure 4B: the Nomarsky (visible) image must be shown.
Figure 5: the appropriate study should have been to compare with 0,125 of HSP27 plus 0,125 of FAIM, not the double amount.
The conclusion is interesting but highly speculative. It is out of interest to characterized the amino acid sequence targeted by FAIM.
Author Response
Reviewer 1
Comments and Suggestions for Authors
In this manuscript the authors try to demonstrate a physical interaction between HSP27 and FAIM in preventing abnormal aggregation of protein upon heat stress.
However, it is not clear what small HSPs the authors talk about, because in there experiment they were using either a/b crystallin, HSP27, HSBP5. Hence, the Title: “Additional effect” should be used instead of “synergic” because small HSPs and FAIM have probably target different substrates.
A) We appreciate the reviewer’s suggestion. We agree that sHSP and FAIM might target different substrates. We have replaced ‘’synergic’’ with ‘’additional’’ and ‘’synergize’’ with ‘’collaborate’’ throughout the manuscript, including the title.
Questions:
Is HSP27 like FAIM block de novo formation of aggregates upon HS?
A) According to the published article (PMID:27909051, Figure 2B and 2C, and PMID: 25699602Figure 5B and 5C), HSP27 can block de novo formation of protein aggregates upon HS. We appreciate the reviewer’s calling this to our attention. This is another similarity between HSP27 and FAIM. We have added ‘’similar to HSP27’’ in the first sentence of the Discussion section, and cited the papers accordingly in order to strengthen the similarity.
The sequence alignment of Hsp25 between C. elegans and human should be shown in supplementary.
A) The reviewer suggested the sequence alignment of Hsp25 between C. elegans and human should be included. We appreciate this suggestion and have added the alignment in the new Supplementary Figure 1 and added some words in the Introduction.
Is FAIM upregulated in C. elegans upon stress?
A) According to the published literature, FAIM is not upregulated in elegans upon stress by Microarray (PMID: 11997337 and 29228038) and RNAseq (PMID: 27496166) analyses. Actually, heat shock elements (HSE) are not observed in the C. elegans faim locus. We conclude that FAIM expression in general is not upregulated during the course of HS.
Nomenclature must be unified HSBP5 = HSP27? small HSPs…
A) We apologize for this mistake. We have updated the texts in Figure 1 and the Figure Legend 1. We also modified the Results section when they appear in the manuscript for the first time to avoid the confusion as follows:
HSP27 (also termed HSPB1).
αA-crystallin (also termed HSPB4).
αB-crystallin (also termed HSPB5).
Figure 2: Why FAIM aggregates disappeared at 2h upon HS, is there re-solubilization of FAIM at 2H-R18 upon heat shock or lost upon cell division?
A) We appreciate the question. We speculate that insoluble FAIM protein is becoming soluble during the recovery period after incubating cells at 37°C because it is too fast for cells to divide during the recovery period. However, we are not sure whether FAIM is solubilized by other chaperons. Rather, we hypothesize that FAIM binds insoluble ubiquitinated proteins upon HS and then is released from the complex during the recovery period. Similarly, it was previously shown by our group that insoluble protein aggregates were also solubilized at 2H-R18 (PMID: 32175331, Figure 3A), and that FAIM is recruited to a complex containing ubiquitinated protein upon HS (PMID: 32175331, Figure 2).
Figure 3 and supplementary Fig 3: is there an explanation why the two figures show a WB with a different distribution of HSP27?
A) We appreciate the reviewer’s calling this to our attention. According to the literature, the expression and cellular distribution of HSP27 and its homologs are cell-specific presumably due to their phosphorylation status, oligomerization status and co-factors (PMID: 7673331, 8168520, 878182, 9378751, 16733662, and 27756555). Therefore, it is not surprising that the cellular distribution of HSP27 is different between HeLa (Figure 3) and HLE-B3 (supplementary Figure 3) cells. The main point of these data is that FAIM and HSP27 translocated into the same fraction (cytoskeletal/insoluble fraction) upon HS.
Figure 4A (and supp. Fig 2) IP: Why there is a decrease in HSP27-FLAG-FAIM interaction at 30 min at 43°C, while FAIM is still aggregated after 2H at 43°C in the Figure 1?
A) We thank the reviewer for the insightful comment. As observe in Figure 4A (2% total), the total amount of soluble FLAG-FAIM is decreased upon HS presumably due to a partial shift to the insoluble fraction. We speculate that the decrease in HSP27-FLAG-FAIM interaction might be caused by the decreased amounts of FLAG-FAIM in the soluble fraction.
The utilization of menadione (a ROS inducer) is not explained nor justified.
A) We appreciate the reviewer’s calling this to our attention. We have added the words, ‘’upon heat shock and oxidative stress induced by menadione’’ in the Results section to explain the utilization of menadione.
HSP27 is highly expressed but only a small fraction is coprecipitated with FAIM-FLAG. This may be relevant of an indirect interaction (via some common targets).
A) We appreciate the insightful comment. At this stage, we do not know if this interaction is direct or indirect. This is why we say ‘’FAIM is recruited to the complex containing sHSPs after cellular stress induction’’ throughout the manuscript instead of, ’’FAIM binds sHSPs after cellular stress induction’’.
Figure 5: the appropriate study should have been to compare with 0,125 of HSP27 plus 0,125 of FAIM, not the double amount.
A) We appreciate the insightful comment. Actually, we tried 125 μM of HSP27 plus 0.125 μM of FAIM but did not observe the synergistic effect in these conditions. Because we agree with the reviewer’s suggestion above, we have replaced ‘’synergic’’ with ‘’additional’’ and ‘’synergize’’ with ‘’collaborate’’ throughout the manuscript.
The conclusion is interesting but highly speculative. It is out of interest to characterized the amino acid sequence targeted by FAIM.
A) We thank the reviewer for the insightful comment. We agree that this is an important avenue to pursue and we have not started along this path. However, the present manuscript concerns the collaboration between FAIM and HSP27 to prevent protein aggregation. We feel that our current work on collaboration makes a complete story that should be communicated without the delay involved in generating additional data on the screening for FAIM’s clients/targets. Moreover, the characterization of the amino acid sequence targeted by FAIM is a somewhat different issue and needs robust work. The results will most likely depend on cell types and could lead to the conclusion that FAIM’s targets are promiscuous. For these reasons, we prefer to report our results on the collaboration between sHSPs and FAIM to prevent protein aggregation without adding potentially interesting data on the targets/clients of FAIM. However, this very interesting, important issue will be addressed in future work.

Reviewer 2 Report
In the manuscript the authors presented results considering properties of FAIM and sHSPs interaction. This manuscript is the continuation of the articles previously published by the authors. In their work the authors elucidate different aspects of FAIM functions: they declare that FAIM may interact with sHSPs (HSP27 as well as αA- and αB-crystallins) to intensify their folding properties. The perspective of the work is in vivo studies. Besides, neuronal cell cultures would be useful to estimate the efficiency of FAIM and sHSPs in prevention of accumulation of misfolded protein aggregates.
The minor comment:
According to requirements of the IJMS, in the text, reference numbers should be placed in square brackets [ ], and placed before the punctuation; for example [1], [1–3] or [1,3]. Thus, the authors should fortmat references in the text as well as references in the reference list in accordance to IJMS requirements.
Author Response
Reviewer 2
Comments and Suggestions for Authors
In the manuscript the authors presented results considering properties of FAIM and sHSPs interaction. This manuscript is the continuation of the articles previously published by the authors. In their work the authors elucidate different aspects of FAIM functions: they declare that FAIM may interact with sHSPs (HSP27 as well as αA- and αB-crystallins) to intensify their folding properties. The perspective of the work is in vivo studies. Besides, neuronal cell cultures would be useful to estimate the efficiency of FAIM and sHSPs in prevention of accumulation of misfolded protein aggregates.
A) We appreciate the comment. Actually, we tried to establish FAIM- and sHSP- doubly overexpressing cell lines using HeLa, and SH-SY5Y cells for this purpose, but failed to obtain them for unknown reasons. We transfected pcDNA3.3 plasmids containing human FAIM or sHSPs genes into these cells, and screened overexpressing cells using G418 but did not obtain any doubly-overexpressing clone. We have concluded that non-physiological overexpression of both FAIM and sHSPs might have some unknown, cytotoxic effects on cells.
The minor comment:
According to requirements of the IJMS, in the text, reference numbers should be placed in square brackets [ ], and placed before the punctuation; for example [1], [1–3] or [1,3]. Thus, the authors should fortmat references in the text as well as references in the reference list in accordance to IJMS requirements.
A) We thank the reviewer for pointing out the wrong format. We apologize for this mistake. We have updated the format of the reference list accordingly.

Reviewer 3 Report
The authors showed that the cytoprotective Fas Apoptosis Inhibitory Molecule (FAIM) shifts to a detergent (RIPA buffer)-insoluble fraction in response to heat shock, as does HSP27, and FAIM appears to associate with HSP27 with stress. On the whole the work is rather preliminary, with the data lacking in multiple aspects.
Major issues:
11. The authors’ principle method of stress induction is heat shock at 43oC. One wonders if other forms of stress (such as calcium ionophores, H2O2, nutrient starvation, apoptosis inducers etc) would produce the same phenomena with FAIM and HSP27. To use the word stress through the manuscript as the authors have done, these other stressors should at least be checked.
22. The problem with heat shock at 43oC is that this temperature sufficient to denature some, if not many proteins, reversibly or irreversibly. Precipitation from solution as a result of denaturation is not unexpected, and not particularly meaningful. Also, heat shock proteins binding to denatured proteins is only to be expected. Do the authors have evidence that FAIM did not simply denature after heat shock at 43oC?
33. The authors showed that FAIM transcript levels did not increase with heat shock, unlike the other HSPs, in Fig 1. However, conspicuously missing from Fig 1 is data for HSP27. In this regard, why did the authors only looked at HSP27 and αA/αB-crystallins but not the other sHSPs? The scope of investigation should indeed be broader here.
44. In Fig 3, the authors showed that FAIM shifts to a vimentin positive, cytoskeletal/detergent-insoluble fraction with a somewhat crude scheme of subcellular fractionation. Does FAIM actually associate with the cytoskeleton with heat shock? This should be demonstrated by immunofluorescence microscopy.
55. The authors attempted to demonstrate FAIM’s association with HSP27 upon heat shock with co-immunoprecipitation of over-expressed tagged proteins after transient transfection. The results based on over-expression are not particularly reliable. The authors should investigate the fate and interactions of the endogenous FAIM.
66. Under what conditions and in which cellular compartments does FAIM interact with HSP27? The authors should check if there are soluble FAIM-HSP27 complexes that could be detected by gel filtration or density gradient sedimentation, and study how these complex(es) change upon induction.
77. The authors appeared to use menadione as an oxidative stressor in experiments in Fig 4. Is the effect reversible by an anti-oxidant or ROS quencher?
88. The proximity ligation results in Fig 4B,C is a little difficult to understand as they are presented. Are the authors showing proximity ligation for FAIM and HSP27? How did the red fluorescence labeling come about and how does that demonstrate proximity between FAIM and HSP27?
99. The authors showed that recombinant FAIM suppresses amyloid-β fibrillization/aggregation in the conjunction with HSP27 in the test tube. Is this due to an interaction between the FAIM with amyloid-β oligomers? If so this interaction should be shown, and the dependence of this interaction on HSP27 should be properly titrated. Do the interaction occur in cells (or neurons)? This should be investigated by co-IP or pull-down experiments.
110. Could the authors demonstrate that FAIM’s association with HSP27 is of some physiological or pathophysiological relevance in other ways? As mentioned above, association of sHSPs with denatured proteins is only to be expected. How does FAIM actually function in protection against cell death if it simply sequester away HSP27 during stress? Would FAIM function/activity actually diminish with a loss of HSP27 by gene silencing or knockout?
Other issues:
The western blots have no molecular size markings and the micrographs have no scale bars.
Author Response
Reviewer 3
Comments and Suggestions for Authors
The authors showed that the cytoprotective Fas Apoptosis Inhibitory Molecule (FAIM) shifts to a detergent (RIPA buffer)-insoluble fraction in response to heat shock, as does HSP27, and FAIM appears to associate with HSP27 with stress. On the whole the work is rather preliminary, with the data lacking in multiple aspects.
Major issues:
1. The authors’ principle method of stress induction is heat shock at 43oC. One wonders if other forms of stress (such as calcium ionophores, H2O2, nutrient starvation, apoptosis inducers etc) would produce the same phenomena with FAIM and HSP27. To use the word stress through the manuscript as the authors have done, these other stressors should at least be checked.
A) We apologize for the confusion. We actually used one of the other forms of stress, oxidative stress inducers, using menadione (Figures 2, and 4, and Supplementary Figure 4 in the revised version of the manuscript). As the Reviewer #1 also mentioned, we have added the words, ‘’upon heat shock and oxidative stress induced by menadione’’ in the Results section to explain the utilization of menadione.
2. The problem with heat shock at 43oC is that this temperature sufficient to denature some, if not many proteins, reversibly or irreversibly. Precipitation from solution as a result of denaturation is not unexpected, and not particularly meaningful. Also, heat shock proteins binding to denatured proteins is only to be expected. Do the authors have evidence that FAIM did not simply denature after heat shock at 43oC?
A) We appreciate the insightful comment. We agree that there might be a possibility that FAIM can be denatured upon HS, resulting in the formation of the FAIM precipitation, which we observed as an insoluble protein (Figure 2). We performed a new experiment in which HeLa cell lysates and purified recombinant FAIM protein in addition to HeLa cells were incubated at 43°C. FAIM protein from HeLa cells lysates and recombinant FAIM protein stayed in the detergent soluble fraction, which strongly suggests that FAIM protein itself is resistant to denaturation caused by high temperature and that it is actively translocated into the insoluble fraction in the live cells. Recombinant FAIM protein is still soluble even after the incubation at 70°C, although soluble multimer FAIM protein bands were observed. Because this is a very important observation in order to exclude the possibility that FAIM precipitates upon HS, we include the data in the new Supplementary Figure 2. We have revised the Result section accordingly.
3. The authors showed that FAIM transcript levels did not increase with heat shock, unlike the other HSPs, in Fig 1. However, conspicuously missing from Fig 1 is data for HSP27. In this regard, why did the authors only looked at HSP27 and αA/αB-crystallins but not the other sHSPs? The scope of investigation should indeed be broader here.
A) We thank the reviewer for raising this point. We have added the HSP27 mRNA data in Figure 1. HSP27 mRNA is slightly upregulated upon HS in HeLa cells. Regarding the protein expression of other human sHSPs (HSPB2, HSPB3, HSPB6, HSPB7, HSPB8, HSPB9, and HSPB10), we understand that all of sHSPs expression should be ideally analyzed. However, we failed to observe their expression in HeLa and HLE B-3 cells by Western Blot. As described in the manuscript, we first identified the similarity between FAIM and HSP27 using HeLa cells (Figure 3), and then confirmed/expanded to αA/αB-crystallinsusing HLE B-3 cells (Supplementary Figure 2). We feel that showing three sHSPs expression should be enough to demonstrate the similar behavior. Because the sHSPs expression pattern depends on cell types (especially, HSPB9 and HSPB10 are expressed in only testicular cells), it is more challenging to show all of their expression.
4. In Fig 3, the authors showed that FAIM shifts to a vimentin positive, cytoskeletal/detergent-insoluble fraction with a somewhat crude scheme of subcellular fractionation. Does FAIM actually associate with the cytoskeleton with heat shock? This should be demonstrated by immunofluorescence microscopy.
A) We appreciate the comment. Actually, we initially hypothesized that FAIM might be associated with cytoskeletal proteins upon HS after we observed the shift. Then, we performed some experiments using confocal microscopy to examine whether FAIM is co-localized with F-actin, tubulin, vimentin, or keratin filaments after HS using HeLa. We, however, found negative results, suggesting that FAIM is not recruited to cytoskeletal proteins upon HS, although more experiments are required to prove the negative evidence. Furthermore, we found that p62 (also termed SQSTM1), which binds ubiquitinated protein aggregates and is involved in the autophagic pathway, is also observed in the cytoskeletal/detergent-insoluble fraction, suggesting that the fraction contains both cytoskeletal proteins and proteins that bind insoluble protein aggregates.
5. The authors attempted to demonstrate FAIM’s association with HSP27 upon heat shock with co-immunoprecipitation of over-expressed tagged proteins after transient transfection. The results based on over-expression are not particularly reliable. The authors should investigate the fate and interactions of the endogenous FAIM.
A) We appreciate the comment. We agree that we should confirm the association between the endogenous FAIM and endogenous sHSPs to exclude the possibility that the association is observed due to the overexpression of FAIM. Although we generated some anti-FAIM antibodies in-house and used commercially available antibodies, we have never found useful antibodies for FAIM immunoprecipitation presumably due to a high homology of FAIM proteins among species. As far as we know, useful antibodies for FAIM immunoprecipitation have not been established (or published). Due to this issue, we ended up using the FLAG-tag-FAIM proteins for co-immunoprecipitation experiments.
6. Under what conditions and in which cellular compartments does FAIM interact with HSP27? The authors should check if there are soluble FAIM-HSP27 complexes that could be detected by gel filtration or density gradient sedimentation, and study how these complex(es) change upon induction.
A) We apologize for the confusion. For all of the shown co-immunoprecipitation experiments, we only used the detergent-soluble fraction based on a general immunoprecipitation protocol. We spun the lysates after cell lysis and added the primary antibody to the supernatant. These procedures are described in the Materials and Methods section. Therefore, the observed complex is in the soluble fraction, suggesting that FAIM is recruited to a complex containing sHSPs before they become insoluble. We performed in situ PLA (Figure 4B) in order to examine in which cellular compartments FAIM interacts with HSP27. However, we did not observe any particular intracellular pattern of the association, suggesting that the association occurs in any cellular compartment as long as cellular stress is induced.
- The authors appeared to use menadione as an oxidative stressor in experiments in Fig 4. Is the effect reversible by an anti-oxidant or ROS quencher?
A) We appreciate the reviewer bringing this to our attention, but we have not explored this potential mechanism at this time. It is out of focus in the current work whether the association is ROS-dependent or ROS-independent although this is an insightful comment. We will look to future study for an answer.
8. The proximity ligation results in Fig 4B,C is a little difficult to understand as they are presented. Are the authors showing proximity ligation for FAIM and HSP27? How did the red fluorescence labeling come about and how does that demonstrate proximity between FAIM and HSP27?
A) We apologize for the confusion as to the principle of PLA. We have explained more about the PLA experiments in the Materials and Methods section to clarify the detailed procedures and the principle of PLA.
9. The authors showed that recombinant FAIM suppresses amyloid-β fibrillization/aggregation in the conjunction with HSP27 in the test tube. Is this due to an interaction between the FAIM with amyloid-β oligomers? If so this interaction should be shown, and the dependence of this interaction on HSP27 should be properly titrated. Do the interaction occur in cells (or neurons)? This should be investigated by co-IP or pull-down experiments.
A) We thank the reviewer for raising this point. Following the recommendations, we have successfully purified amyloid-β oligomers by a size exclusion column and performed Co-IP experiments using 2 anti-amyloid-β antibodies (clone D54D2 from Cell Signaling Technology and clone 6E10 from BioLegend). However, we encountered some technical issues. Although both clones were able to pull-down amyloid-β as previously described, they are cross-reactive with FAIM. The antibodies can also pull-down FAIM even without amyloid-β. In spite of this issue, we will investigate the mechanism using different methods in the future.
- Could the authors demonstrate that FAIM’s association with HSP27 is of some physiological or pathophysiological relevance in other ways? As mentioned above, association of sHSPs with denatured proteins is only to be expected. How does FAIM actually function in protection against cell death if it simply sequester away HSP27 during stress? Would FAIM function/activity actually diminish with a loss of HSP27 by gene silencing or knockout?
A) As mentioned above (comment 2), we excluded the possibility that FAIM becomes insoluble upon HS due to the denaturation of FAIM protein caused by high temperature. The data also excludes the possibility that HSP27 simply interacts with denatured FAIM upon HS. We tried to obtain HSP27-deficient HeLa cells, but failed to obtain them for unknown reasons. We transfected PX458 or PX330 plasmid expressing human HSP27 guide RNA and Cas9 protein into HeLa cells, and screened HSP27-deficient cells after limiting dilution but did not obtain any HSP27-deficient clone. We speculate that HSP27 might be involved in the survival of HeLa cells.
Other issues:
The western blots have no molecular size markings and the micrographs have no scale bars.
A) We apologize for this inconvenience. We have added molecular size markers and scale bars.
Round 2
Reviewer 3 Report
1. The authors’ principle method of stress induction is heat shock at 43oC. One wonders if other forms of stress (such as calcium ionophores, H2O2, nutrient starvation, apoptosis inducers etc) would produce the same phenomena with FAIM and HSP27. To use the word stress through the manuscript as the authors have done, these other stressors should at least be checked.
A) We apologize for the confusion. We actually used one of the other forms of stress, oxidative stress inducers, using menadione (Figures 2, and 4, and Supplementary Figure 4 in the revised version of the manuscript). As the Reviewer #1 also mentioned, we have added the words, ‘’upon heat shock and oxidative stress induced by menadione’’ in the Results section to explain the utilization of menadione.
2nd review: The use of menadione is already noted in the previous review, but this is only for a small part of the paper. A majority of the paper uses heat shock at 43oC, which denatures proteins. The bottom line is this: the authors must show that their observed phenomenon is a result of an intrinsic physiological stress response and not due non-specifically to external physical or chemical factors. The authors should try a range of stressors, including cell death and ER stress inducing agents, and document the relevant effects of these,
2. The problem with heat shock at 43oC is that this temperature sufficient to denature some, if not many proteins, reversibly or irreversibly. Precipitation from solution as a result of denaturation is not unexpected, and not particularly meaningful. Also, heat shock proteins binding to denatured proteins is only to be expected. Do the authors have evidence that FAIM did not simply denature after heat shock at 43oC?
A) We appreciate the insightful comment. We agree that there might be a possibility that FAIM can be denatured upon HS, resulting in the formation of the FAIM precipitation, which we observed as an insoluble protein (Figure 2). We performed a new experiment in which HeLa cell lysates and purified recombinant FAIM protein in addition to HeLa cells were incubated at 43°C. FAIM protein from HeLa cells lysates and recombinant FAIM protein stayed in the detergent soluble fraction, which strongly suggests that FAIM protein itself is resistant to denaturation caused by high temperature and that it is actively translocated into the insoluble fraction in the live cells. Recombinant FAIM protein is still soluble even after the incubation at 70°C, although soluble multimer FAIM protein bands were observed. Because this is a very important observation in order to exclude the possibility that FAIM precipitates upon HS, we include the data in the new Supplementary Figure 2. We have revised the Result section accordingly.
2nd review: These are notable improvements. However, being detergent soluble does not mean that a protein is not denatured. After all, SDS-denatured protein is soluble in buffer containing SDS. Denaturation or otherwise can only be properly shown by biophysical methods, or by function. Again, the bottom line is that the onus is upon the authors to show that under stress, FAIM does not denature and that FAIM’s recruitment to sHSP-containing complexes upon stress is a physiological stress response.
3. The authors showed that FAIM transcript levels did not increase with heat shock, unlike the other HSPs, in Fig 1. However, conspicuously missing from Fig 1 is data for HSP27. In this regard, why did the authors only looked at HSP27 and αA/αB-crystallins but not the other sHSPs? The scope of investigation should indeed be broader here.
A) We thank the reviewer for raising this point. We have added the HSP27 mRNA data in Figure 1. HSP27 mRNA is slightly upregulated upon HS in HeLa cells. Regarding the protein expression of other human sHSPs (HSPB2, HSPB3, HSPB6, HSPB7, HSPB8, HSPB9, and HSPB10), we understand that all of sHSPs expression should be ideally analyzed. However, we failed to observe their expression in HeLa and HLE B-3 cells by Western Blot. As described in the manuscript, we first identified the similarity between FAIM and HSP27 using HeLa cells (Figure 3), and then confirmed/expanded to αA/αB-crystallinsusing HLE B-3 cells (Supplementary Figure 2). We feel that showing three sHSPs expression should be enough to demonstrate the similar behavior. Because the sHSPs expression pattern depends on cell types (especially, HSPB9 and HSPB10 are expressed in only testicular cells), it is more challenging to show all of their expression.
2nd review: The reason for checking the expressions of HSPs is to fully determine, or differentiate, between the FAIM-associated phenomena from that of a canonical heat-shock or stress response, as for the latter the HSP transcripts would be elevated. If FAIM is not a heat-shock response protein, then the phenomenon of it being recruited into detergent insoluble complexes with HSPs may not be part of a heat-shock response, and the physiological significance of this phenomenon remains very much unclear.
4. In Fig 3, the authors showed that FAIM shifts to a vimentin positive, cytoskeletal/detergent-insoluble fraction with a somewhat crude scheme of subcellular fractionation. Does FAIM actually associate with the cytoskeleton with heat shock? This should be demonstrated by immunofluorescence microscopy.
A) We appreciate the comment. Actually, we initially hypothesized that FAIM might be associated with cytoskeletal proteins upon HS after we observed the shift. Then, we performed some experiments using confocal microscopy to examine whether FAIM is co-localized with F-actin, tubulin, vimentin, or keratin filaments after HS using HeLa. We, however, found negative results, suggesting that FAIM is not recruited to cytoskeletal proteins upon HS, although more experiments are required to prove the negative evidence. Furthermore, we found that p62 (also termed SQSTM1), which binds ubiquitinated protein aggregates and is involved in the autophagic pathway, is also observed in the cytoskeletal/detergent-insoluble fraction, suggesting that the fraction contains both cytoskeletal proteins and proteins that bind insoluble protein aggregates.
2nd review: The elaboration above does little to indicate that FAIM insolubility and HSP interaction with heat-shock is of physiological significance, and in fact promoted the suspicion that we are looking at a complex aggregation of misfolded protein made ready for clearance from the cell. Could the authors see if FAIM gets incorporated into autophagic structures or are cleared eventually by the autophagic flux?
5. The authors attempted to demonstrate FAIM’s association with HSP27 upon heat shock with co-immunoprecipitation of over-expressed tagged proteins after transient transfection. The results based on over-expression are not particularly reliable. The authors should investigate the fate and interactions of the endogenous FAIM.
A) We appreciate the comment. We agree that we should confirm the association between the endogenous FAIM and endogenous sHSPs to exclude the possibility that the association is observed due to the overexpression of FAIM. Although we generated some anti-FAIM antibodies in-house and used commercially available antibodies, we have never found useful antibodies for FAIM immunoprecipitation presumably due to a high homology of FAIM proteins among species. As far as we know, useful antibodies for FAIM immunoprecipitation have not been established (or published). Due to this issue, we ended up using the FLAG-tag-FAIM proteins for co-immunoprecipitation experiments.
2nd review: The authors explained their resource limitations, but that does not help to clarify the nature and mode of FAIM-HSP27 interaction, as to whether these occur between fully folded and functional FAIM (which could be attested by FAIM antibodies) or simply misfolded or denature FAIM (which termini-tagged short epitopes would still work in IP).
6. Under what conditions and in which cellular compartments does FAIM interact with HSP27? The authors should check if there are soluble FAIM-HSP27 complexes that could be detected by gel filtration or density gradient sedimentation, and study how these complex(es) change upon induction.
A) We apologize for the confusion. For all of the shown co-immunoprecipitation experiments, we only used the detergent-soluble fraction based on a general immunoprecipitation protocol. We spun the lysates after cell lysis and added the primary antibody to the supernatant. These procedures are described in the Materials and Methods section. Therefore, the observed complex is in the soluble fraction, suggesting that FAIM is recruited to a complex containing sHSPs before they become insoluble. We performed in situ PLA (Figure 4B) in order to examine in which cellular compartments FAIM interacts with HSP27. However, we did not observe any particular intracellular pattern of the association, suggesting that the association occurs in any cellular compartment as long as cellular stress is induced.
2nd review: That FAIM “…is recruited to a complex containing sHSPs before they become insoluble” is not particularly surprising, not is on its own interesting. Upon heat-shock, denatured and misfolded FAIM would be surrounded by the elevated HSPs and could remain soluble until further their aggregation into insoluble forms. This point needs further verification. Could different molecular sizes of soluble and insoluble FAIM-HSP27 complexes be shown? Importantly, why do FAIM in stressed cells bind HSP27 and what physiological function(s) does this binding serve?
7. The authors appeared to use menadione as an oxidative stressor in experiments in Fig 4. Is the effect reversible by an anti-oxidant or ROS quencher?
A) We appreciate the reviewer bringing this to our attention, but we have not explored this potential mechanism at this time. It is out of focus in the current work whether the association is ROS-dependent or ROS-independent although this is an insightful comment. We will look to future study for an answer.
2nd review: The authors’ answer does not help. Showing whether the stress process is reversible is important to determine its physiological nature. If something is specifically dependent on oxidative stress, quenching that stress should reverse it, for otherwise the chance of something non-specific occurring is higher. Further to the above, it would also be pertinent to ask if the heat-shock induce phenomenon could be reverse. In other words, would the detergent insolubility of FAIM or its interaction with HSPs resolve over time with removal of the heat-shock or oxidative stress?
8. The proximity ligation results in Fig 4B,C is a little difficult to understand as they are presented. Are the authors showing proximity ligation for FAIM and HSP27? How did the red fluorescence labeling come about and how does that demonstrate proximity between FAIM and HSP27?
A) We apologize for the confusion as to the principle of PLA. We have explained more about the PLA experiments in the Materials and Methods section to clarify the detailed procedures and the principle of PLA.
9. The authors showed that recombinant FAIM suppresses amyloid-β fibrillization/aggregation in the conjunction with HSP27 in the test tube. Is this due to an interaction between the FAIM with amyloid-β oligomers? If so this interaction should be shown, and the dependence of this interaction on HSP27 should be properly titrated. Do the interaction occur in cells (or neurons)? This should be investigated by co-IP or pull-down experiments.
A) We thank the reviewer for raising this point. Following the recommendations, we have successfully purified amyloid-β oligomers by a size exclusion column and performed Co-IP experiments using 2 anti-amyloid-β antibodies (clone D54D2 from Cell Signaling Technology and clone 6E10 from BioLegend). However, we encountered some technical issues. Although both clones were able to pull-down amyloid-β as previously described, they are cross-reactive with FAIM. The antibodies can also pull-down FAIM even without amyloid-β. In spite of this issue, we will investigate the mechanism using different methods in the future.
2nd review: The authors’ explanation does not help. Nothing is shown and we still do not know if FAIM could really interact with amyloid-β oligomers, whether this interaction is HSP27 dependent or inhibited, or whether this could occur in cells.
10. Could the authors demonstrate that FAIM’s association with HSP27 is of some physiological or pathophysiological relevance in other ways? As mentioned above, association of sHSPs with denatured proteins is only to be expected. How does FAIM actually function in protection against cell death if it simply sequester away HSP27 during stress? Would FAIM function/activity actually diminish with a loss of HSP27 by gene silencing or knockout?
A) As mentioned above (comment 2), we excluded the possibility that FAIM becomes insoluble upon HS due to the denaturation of FAIM protein caused by high temperature. The data also excludes the possibility that HSP27 simply interacts with denatured FAIM upon HS. We tried to obtain HSP27-deficient HeLa cells, but failed to obtain them for unknown reasons. We transfected PX458 or PX330 plasmid expressing human HSP27 guide RNA and Cas9 protein into HeLa cells, and screened HSP27-deficient cells after limiting dilution but did not obtain any HSP27-deficient clone. We speculate that HSP27 might be involved in the survival of HeLa cells.
2nd review: As explained in point 2 above, the authors did not, as they claimed, exclude the possibility that FAIM becomes denatured or misfolded with heat-shock, nor that HSP27 simply interacts with denatured FAIM. The key question here of how FAIM actually function in protection against cell death if it simply sequester away HSP27 during stress is not answered. Also, whether FAIM’s protective activity is dependent on its association with HSPs is not answered.
In summary, the results did not show convincingly that FAIM's binding to HSP27 in a detergent-insoluble complex upon heat-shock is
a) not simply due to FAIM being denature or misfolded,
b) part of a cellular heat-shock or stress response program, pathway or process that serves an recognizable physiological function, and
c) mechanistically related to FAIM's known neuroprotective property.
Author Response
We appreciate the reviewer’s comment. The reviewer speculates that “FAIM becomes denatured or misfolded with heat-shock” and that “HSP27 simply interacts with denatured FAIM” as an explanation for its appearance with HSP27 in a detergent-insoluble complex. This seems highly unlikely on the basis of results previously submitted in response to the first round review. We call attention to Supplementary Figure 2A that shows that FAIM does NOT move to the detergent-insoluble complex when HeLa cell lysates are heat shocked, as would be expected for a protein that becomes denatured upon heat shock and for that reason moves to the detergent-insoluble complex. Thus, in light of this result, it seems untenable to propose (as in part a of the reviewer’s comments) that heat-induced translocation of FAIM to the detergent insoluble complex is due to denaturation of FAIM.
Regarding FAIM binding to/by HSP27, we apologize for any confusion in the previous letter and manuscript. Actually, we do not say ‘’binding’’ throughout the manuscript. Instead, we say ‘’FAIM is recruited to HSP27-containing complex’’. We do not say FAIM- HSP27 association happens in the detergent-insoluble fraction simply because co- immunoprecipitation can be performed using only detergent-soluble proteins as already described in the Methods section. Furthermore, as explained above and as mentioned in the letter for the first round of the review cycle, we showed that FAIM is recruited to HSP27-containing complex in a live-cell intrinsic manner, and that FAIM protein is resistant to heat shock. These observations strongly suggest that the association of FAIM and HSP27 upon heat shock is not simply due to FAIM being denatured or misfolded. Thus, we hypothesize that both FAIM and HSP27 are recruited to the same complex containing ubiquitinated protein for degradation upon heat shock and work together (Fig 5) to prevent protein aggregation more efficiently as we previously showed that FAIM is recruited to the complex containing ubiquitinated protein upon heat shock or oxidative stress. This hypothesis is newly added in the Discussion section in order to make the point of the manuscript clearer.
Regarding part b of the reviewer’s comments, we do not say that the collaborative function of FAIM and HSP27 outlined in our work indicates a recognizable physiological function. We have presented a series of experiments/results conducted in vitro with well- recognized stress triggers and well-known cell lines, to elucidate novel molecular interactions. This is a well-worn path to discovery. he reviewer’s comment is important, and we plan further work to determine the physiological role of FAIM in primary cells (which is suggested by our previous results on artificially stressed primary fibroblasts and whole animals).
Regarding part c of the reviewer’s comments, we have not published any role for FAIM in neuroprotection up to now. Therefore, from our standpoint and based on our data, FAIM's neuroprotective property is ‘’UNKNOWN’’. The reviewer’s comment is important, and we plan to directly determine FAIM activity in neurons in future work.
We hope this clarifies our work. We appreciate the reviewers’ comments that have helped improve our work and hope that our findings are now fully acceptable for publication.
